# Associated Factors of High Sedative Requirements within Patients with Moderate to Severe COVID-19 ARDS

**DOI:** 10.3390/jcm11030588

**Published:** 2022-01-25

**Authors:** Armin N. Flinspach, Hendrik Booke, Kai Zacharowski, Ümniye Balaban, Eva Herrmann, Elisabeth H. Adam

**Affiliations:** 1Department of Anaesthesiology, Intensive Care Medicine and Pain Therapy, Goethe-University Frankfurt, 60590 Frankfurt/Main, Germany; j.booke@icloud.com (H.B.); kai.zacharowski@kgu.de (K.Z.); Elisabeth.Adam@kgu.de (E.H.A.); 2Department of Biostatistics and Mathematical Modelling, Goethe-University Frankfurt, 60590 Frankfurt/Main, Germany; Balaban@med.uni-frankfurt.de (Ü.B.); herrmann@med.uni-frankfurt.de (E.H.)

**Keywords:** critical care, hypnotics and sedatives, acute respiratory distress syndrome, severe acute respiratory syndrome coronavirus 2, pulmonary ventilation, prone position

## Abstract

The coronavirus pandemic continues to challenge global healthcare. Severely affected patients are often in need of high doses of analgesics and sedatives. The latter was studied in critically ill coronavirus disease 2019 (COVID-19) patients in this prospective monocentric analysis. COVID-19 acute respiratory distress syndrome (ARDS) patients admitted between 1 April and 1 December 2020 were enrolled in the study. A statistical analysis of impeded sedation using mixed-effect linear regression models was performed. Overall, 114 patients were enrolled, requiring unusual high levels of sedatives. During 67.9% of the observation period, a combination of sedatives was required in addition to continuous analgesia. During ARDS therapy, 85.1% (*n* = 97) underwent prone positioning. Veno-venous extracorporeal membrane oxygenation (vv-ECMO) was required in 20.2% (*n* = 23) of all patients. vv-ECMO patients showed significantly higher sedation needs (*p* < 0.001). Patients with hepatic (*p* = 0.01) or renal (*p* = 0.01) dysfunction showed significantly lower sedation requirements. Except for patient age (*p* = 0.01), we could not find any significant influence of pre-existing conditions. Age, vv-ECMO therapy and additional organ failure could be demonstrated as factors influencing sedation needs. Young patients and those receiving vv-ECMO usually require increased sedation for intensive care therapy. However, further studies are needed to elucidate the causes and mechanisms of impeded sedation.

## 1. Introduction

The COVID-19 (coronavirus disease 2019) pandemic is probably the greatest health care challenge of the 21st century. Although there is hope that vaccines will reduce the global spread of SARS-CoV-2 (severe acute respiratory syndrome coronavirus 2), infected patients and virus mutations are likely to challenge the health care system for a long period. Approximately 5–10% of infected patients develop COVID-19-induced acute respiratory distress syndrome (CARDS), requiring critical care therapy and mechanical ventilation, including sedation [1,2]. In the ongoing absence of a breakthrough coronavirus-specific treatment, guidelines focus on protective pulmonary ventilation and prone positioning therapy [3,4]. The sedation required for this kind of elaborate critical care therapy in patients with CARDS, including veno-venous extracorporeal membrane oxygenation (vv-ECMO) and prone positioning therapy, has emerged as a sophisticated task [3].

So far, it remains unclear to what extent the various recommendations on sedation strategies published for critically ill patients with acute respiratory distress syndrome (ARDS) are applicable for patients with CARDS [4,5,6,7]. Avoidance of deep sedation while undergoing intensive care treatment is clearly advised in ARDS patients whenever possible [7,8]. This strategy shortens the duration of mechanical ventilation significantly, thus the length of stay in the intensive care unit (ICU), leading to less ICU-related complications. Additionally, otherwise missed cerebrovascular events can be detected more easily [8,9,10]. However, CARDS patients have an exceptionally long treatment duration with the need for repetitive prone positioning and, if necessary, veno-venous extracorporeal membrane oxygenation (vv-ECMO) therapy [3,11]. For these therapies and to achieve sufficient patient–ventilator synchrony, adequate sedation of CARDS patients is required on a regular basis [8,12]. Previous studies demonstrated that critically ill COVID-19 patients pose a substantial challenge regarding sedation management. This was reflected by both unusually high dosages and the frequent need for a combined use of sedatives [13,14]. In addition, various research groups described the use of volatile sedatives to enable patient-adapted therapy [14,15]. These results were also confirmed by a comparison with non-CARDS patients [16].

Nevertheless, the pathophysiological causes and the significance of the underlying mechanisms of difficult sedation in these patients remain unclear. Thus far, there are various hypotheses regarding the cause of this increased sedation requirement among critically ill CARDS patients without support from study/research data [17,18].

Up to now, no study analyzing the influence of therapy regimens and pre-existing conditions has been published. Therefore, in our monocentric study, we aim to assess the underlying factors for the commonly observed increased dosages of analgesic and sedative medications in patients with CARDS. 

## 2. Materials and Methods

We conducted a prospective, observational study at the University Hospital Frankfurt which was approved by the institutional ethics board of the University of Frankfurt (#20-643). A waiver regarding the requirement of written informed consent from COVID-19 patients was authorized. The study was registered to the Clinical Trials.gov Protocol Registration and Results System (NCT04667936). 

This study was planned and designed in accordance with the recommendations of the Strengthening the Reporting of Observational Studies in Epidemiology (STROBE) guidelines, using the suggested checklist [19]. The manuscript adheres to the Consolidated Standards of Reporting Trials (CONSORT) guidelines [20].

### 2.1. Patient Population

We included all patients admitted to our institutional ARDS center-qualified ICU between April 2020 and December 2020 who were already diagnosed with severe acute respiratory syndrome coronavirus type 2 (SARS-CoV-2) infection or tested positive by real-time reverse transcriptase polymerase chain reaction (RT-PCR) during treatment [21].

Sedation was performed according to the in-house sedation protocol (Appendix A); no specific protocol was defined for other treatment modalities, as these were solely at the discretion of the attending physicians.

Study inclusion was based on the presence of ARDS with need for mechanical ventilation and consecutive sedation. Mechanical ventilation was performed by using an Elisa 800 (Löwenstein Medical, Bad Ems, Germany) or a Hamilton G5 (Hamilton Medical, Bonaduz, Switzerland) ICU ventilator, as well as intensive care therapy, according to the current recommendations available for the treatment of CARDS [12,22,23,24].

The observation period began with endotracheal intubation and corresponding sedation or with admission, in patients that had already been intubated. The observation period ended with death, cessation of pharmaceutical sedation after tracheotomy, or successful spontaneous breathing trial and subsequent extubation. In case of persistent oxygenation and/or decarboxylation failure, an interdisciplinary team determined indication for ECMO treatment. This was a patient-specific, case-by-case decision based on available guidelines [25,26]. At the beginning of vv-ECMO therapy, ultraprotective pulmonary ventilation with a tidal volume of ≤6 mL kg^−1^ predicted body weight was targeted, to prevent unnecessary mechanical stress on the lung [27].

In addition to published data (including ours from PLoS ONE) [28] that revealed signs of increased sedation need, we want to investigate whether there are underlying factors that explain these increased sedation dosages.

### 2.2. Data Collection

Clinical data were continuously recorded using a patient data management system (PDMS; Metavision 5.4; iMDsoft, Tel Aviv, Israel). We recorded demographic data, laboratory results, ventilation parameters, sedative dosages, clinical satisfaction of sedation levels, Richmond Agitation and Sedation Scale (RASS), positioning therapy, vv-ECMO therapy and outcomes. 

To exclude temporary requirements of increased sedation, e.g., during an interventional procedure, we only considered sedation regimens with an application lasting more than 4 h for the analyses. 

To determine an adequate level of sedation in addition to the assessment of RASS, there were periodic bedside examinations by the attending physician and intensive care nurse, who assessed ventilatory synchrony, signs of stress and the presence of vegetative agitation. Adequate ventilator synchrony was defined as the clinically predominant absence of asynchronous phases, which was based on the observation of respiratory volume pressure curves by the attending staff [29]. Following published recommendations, a RASS from 0 to −1 was identified as the target of sedation [4]. In prone position and for vv-ECMO therapy, a RASS from −3 to −4 was targeted for sufficient psycho-vegetative protection [9,30]. In prone position, the assessment of the RASS, in terms of eye movement and opening, was carried out by the responsible intensive care staff under limited conditions due to the head being carefully tilted to the side. According to the recommendations, deep sedation for adequate therapy was maintained only for as long as necessary.

Prone positioning was performed within the early available therapy recommendations regarding CARDS therapy [22,23]. The practice of regular prone positioning, which has since been incorporated into the guidelines, was maintained due to the emerging evidence during the course of the pandemic [31,32].

### 2.3. Statistical Analysis

No statistical power calculation was conducted prior to this study. The sample size was based on the available data.

There was a predefined statistical analysis plan. Data with a continuous scale are represented as mean (standard deviation) and data with a categorical scale are presented as frequencies and percentages. Additionally, spontaneous breathing time and RASS values were analyzed using linear regression mixed-effect models with sedation agent-dependent variance calculation.

All statistical tests were two-tailed and results with *p* ≤ 0.05 were considered statistically significant. All calculations/analyses were performed with SPSS (IBM Corp., Version 26, Chicago, IL, USA) or R for Statistical Computing (The R Foundation, Version 4.0, Vienna, Austria). The packages ‘MASS’ and ‘nlme’ were used [33,34].

The selection of appropriate parameters for linear regression was developed by an interdisciplinary team of experienced clinicians and members of the Department of Biostatistics and Mathematical Modelling. The parameters selected represented the most commonly described complications of COVID-19 disease among critically ill patients as well as overall disease severity. These include complete lung failure, renal and hepatic dysfunction and cardiovascular impairment [35].

## 3. Results

We studied 221 patients, 114 of whom met the inclusion criteria and could be included in the analysis (Figure 1).

The clinical and demographic characteristics of the included patients at the time of admission are presented in Table 1. In our study population, no history of drug abuse was detected. Regarding alcohol, two patients had a history of alcohol abuse. However, this was terminated prior to COVID-19 infection. In one patient, the medical history revealed preexisting epilepsy which had been asymptomatic for several years with therapy. In 27 patients, dilatatory tracheostomy was performed. Tracheostomy was performed after a median of 15.5 (IQR, 8.5) days of ventilatory support. Among the 65 (57.0%) CARDS patients who died, 18 received vv-ECMO therapy (vv-ECMO-mortality = 18/23 = 78.2%).

At the time of admission, sixteen patients had a Horovitz (P/F index) index corresponding to mild ARDS, which deteriorated to a Horovitz index <150 in all CARDS patients during the course of intensive care treatment.

In the context of early treatment recommendations for CARDS therapy available at the onset of the pandemic, we performed 847 episodes of prone positioning maneuvers in 97 of the included patients (85.1%). The median prone positioning therapy time was 80.0 (±62.3) h, indicating that patients were in a prone position in 32.9% of the observed study period. A total of 17 patients could not be treated with prone positioning; in 10 of these cases, the patients were in a highly unstable condition and died within the first 2 days of intensive care treatment, while, for the 7 remaining patients, prone positioning was impossible due to super obesity (BMI > 50 kg/cm^2^).

During the course of CARDS therapy, we observed a marked deterioration in oxygenation leading to an increase in ARDS severity according to the BERLIN definition [36]. The majority of patients developed severe *n* = 81 (69.8%) or moderate *n* = 29 (25.0%) ARDS during disease progression. As a result, moderate to severe ARDS was observed in 74.8% of the overall study period.

Adequate sedation for therapy was achieved by administering a continuous single sedation regime in 32.1% of the time. Further, a two-, three- or four-fold sedation was re-quired in 44.4%, 19.8% and 3.7% of the treatment time. We observed a significantly higher total sedative demand (*p* = 0.003) of CARDS patients <65 years.

Increased sedation depth for prone positioning as well as ECMO therapy resulted in a significantly higher sedation requirement (*p* < 0.001). The detailed course of the observed sedation depth (RASS) during treatment is shown in Figure 2. 

Furthermore, a diminishing demand of sedation dosages was found in patients with organ dysfunction of the kidneys (*p* = 0.001). Acute renal failure was defined as ≥stage 2, according to the Kidney Disease Improving Global Outcomes (KDIGO) criteria. In addition, in patients suffering from liver dysfunction (*p* = 0.001), defined as a combination of coagulation disorder and serum bilirubin ≥6.0 mg/dL), increased sedation requirements were found. Additionally, catecholamine therapy was associated with higher sedation dosages (*p* < 0.001). The preexisting conditions we examined had no significant effect on sedation dose consumption (Table 2).

The main sedative agents used were gamma-aminobutyric acid (GABA) receptor active substances propofol (37.9%) with a mean dosage of 2.61 (±1.02) mg·kg^−1^·min^−1^, as well as midazolam (59.9%) with a mean dosage of 0.17 (±0.07) mg·kg^−1^·h^−1^. Clonidine (57.4%) as representative of central α_2_ inhibitors and esketamine (24.5%) as N-methyl-D-aspartate (NMDA) receptor inhibitor with mean dosages of 1.80 (±0.80) µg·kg^−1^·h^−1^ and 1.14 (±0.84) mg·kg^−1^·h^−1^, respectively, were also used. A dosage summary with respect to combined applications is given in Figure 3.

In addition to the application of sedatives, continuous analgesia with potent opioids was given for tube tolerance and for patient-adapted analgesia during prone positioning therapy. In 99.1% of the treatment time, sufentanil was used for analgesia with a mean sufentanil dose of 0.16 (±0.09) µg·kg^−1^·min^−1^. This corresponds to a mean oral morphine equivalent of 706.5 mg·d^−1^ (±374.8) [37,38]. The pharmacokinetically short-acting agents lormetazepam (10.5%) and dexmedetomidine (3.1%), as well as the opioid remifentanil (0.9%) and the volatile sedative sevofluran (1.6%), were used to a lesser extent. Regarding ventilation, we were able to achieve ventilator-assisted spontaneous breathing in 76.52% of the observed treatment period.

During CARDS therapy, four patients required continuous administration of neuromuscular blocking agents (NMBAs). For this purpose, we used the substance cisatracurium (168.0 ± 42.0 mg·d^−1^). In addition, 41 single doses of cisatracurium (mean, 18.0 ± 15.0 mg) and 83 single doses of rocuronium (mean, 90.6 ± 25.0 mg) were used to provide adequate therapy in case of excessive ventilator desynchronization or cough-associated suctioning of vv-ECMO cannulas.

## 4. Discussion

Our prospective observational study data show an analysis of the previously only assumed causes of aggravated sedation. As already shown before, CARDS patients demonstrate an unusually high need for sedation. This also has been shown in the comparison of sedative needs between viral ARDS and CARDS [13,16,28].

Our study data reinforce this increased need for sedatives in terms of dosage and need of their combination. Thus, in addition to high sedative dosages, the need for combined use was found in 2/3 of the recorded treatment period. To achieve the prescribed sedation depth, sedation with midazolam or clondine or the combination of both substances was predominantly used; in addition, 12.8% of CARDS patients showed the need for additional propofol or esketamine application. Our study results are consistent with previously published data in terms of prescription frequency, particularly regarding central α_2_-agonists, benzodiazepines and esketamine. The interaction potential of the sedatives with each other has to be considered when using these substances in combination. Clonidine in particular has considerable co-sedative and co-analgesic potential [39].

The same finding applies to the continuous analgesia we reported, with a median oral morphine equivalent of 659.6 mg/d (±374), which is in line with the results obtained by Kapp et al. [13]. In addition, the relatively high sedative potential of sufentanil should not be neglected, which underlines the complexity of sedation in CARDS patients.

We used the volatile sedative sevoflurane with excellent results in individual cases of particular sedation difficulty, but this has been, so far, reserved for special cases of sedation difficulty [14].

As suggested by colleagues at the beginning of the pandemic, we were able to show that patients of younger age had a significantly higher need for sedation than older patients (>65 years) [3]. This makes the overall younger median age of patients a potential factor for the observed high need for sedation dosages in CARDS patients. Interestingly, according to our data, the patients’ preconditions do not seem to have any influence on the required sedation. At no time during clinical care was there a shortage of sedatives at the treating center that would have necessitated a change in therapy.

Mortality within the study cohort was 57.0%, which may be due to the character of the hospital as a university ARDS center, also with regard to the possibility of ECMO therapy. However, it is also in line with internationally reported mortality rates among critically ill CARDS patients [40]. Among the CARDS patients admitted to our ICU, 20.2% required vv-ECMO therapy due to progressive oxygenation impairment. In a linear regression analysis of the related sedative dosages, we showed that CARDS patients receiving vv-ECMO therapy had a significantly increased sedation requirement (*p* < 0.001) in comparison to patients without vv-ECMO therapy. This finding must be interpreted in light of the fact that the higher dosages found may also be due to the lower target sedation level required to perform vv-ECMO therapy. Additionally, an elimination via the ECMO circuit is a possible reason for the higher observed dosages [8,41,42]. Lipophilic drugs and high protein-bound drugs are prone to sequestration and both are properties of the most frequently used sedatives (e.g., propofol, protein binding at 95–99%) [43]. Therefore, a direct assignment to the distinctive severity of CARDS is limited.

Contrary to the increased need for sedation observed above, we detected reduced sedation dosages in multi-organ failure. Patients with impaired renal function or acute renal failure and liver failure required significantly fewer sedatives according to the linear regression analysis. This may be attributed to the considerable limitation of pharmaceutical metabolism. The administered sedatives undergo hepatic metabolism as well as at least partial renal excretion, which are impaired by the corresponding organ dysfunctions [44,45].

However, our observation of a statistically significant reduction in the required sedatives in the presence of hepatic or renal dysfunction should not be interpreted unquestioningly as CARDS-specific, as corresponding findings have also been observed in organ dysfunction alone [45,46].

With regard to the observed increased need for sedatives in patients with vasoplegia and corresponding catecholamine therapy, a rational conclusion appears to be more difficult. On the one hand, an increased application of sedatives, such as propofol, can independently lead to hypotension, thus to an increased need for catecholamines [47]. On the other hand, it is also feasible that critical organ perfusion requiring catecholamine administration may result in decreased organ perfusion and, consecutively, reduced metabolism which, again, results in reduced need for sedatives [48].

We used NMBAs very cautiously to allow our patients to breathe spontaneously, aiming for improved oxygenation and reduce diaphragmatic muscle atrophy [49,50]. Accordingly, we observed a high rate of ventilator-assisted spontaneous breathing (76.52%) and were able to achieve daily sedation-free intervals. The possible use of NMBAs to improve ventilator synchronization was also highlighted in the work of Wongtangnam et al. with an increase in both the number of NMBAs administered to 56.1% of all CARDS patients treated and the required dosages. Although the use of NMBAs for ARDS therapy is only recommended in the first 48 h after intubation, adequate synchronization without increased use of sedatives or relaxants appears challenging [51,52,53]. In our study population, due to frequent spontaneous breathing, we were mostly able to achieve adequate ventilator synchronization without NMBAs. If this could not be achieved without increasing sedation, we were able to manage this by administering single doses of NMBAs using relaxometry in the early phase of CARDS, with the exception of four cases that required continuous relaxation.

Evidence to explain the initial pathomechanisms of neurocognitive symptoms associated with COVID-19 is increasing [54]. Numerous symptoms of COVID-19 infection, ranging from early hyposmia and ageusia to epileptic seizures and numerous long COVID-19 symptoms, such as fatigue, indicate a substantial influence of COVID-19 infection on neurocognitive function. An association to impaired sedation has yet to be investigated [55]. It has also not yet been possible to clarify the extent to which the increased sedation requirements mentioned above are related to post-intensive care syndrome (PICS). Late sequelae of severe COVID-19 infection observed in the context of PICS include not only physical but also neurological impairment and a reduced quality of life [56,57]. A link via neuroinflammation would be conceivable and should be investigated further.

## 5. Limitations

Our work has some limitations, in particular in terms of study design. We could not obtain a control group from the patients of our hospital to make a direct comparison between COVID-19 ARDS to non-CARDS patients. This was achieved by Wongtangman et al., but they were also limited in establishing a comparable group composition due to the special demographics among COVID-19 patients. Due to the constantly evolving knowledge of the clinical course and treatment of COVID-19 patients, it was not possible to establish a treatment protocol. At our center, we established a treatment algorithm to provide critical care therapy as put forth by Poston et al. This treatment algorithm has been repeatedly adjusted based on new scientific knowledge [58].

As a university center, we had a special mandate to care for critically ill COVID-19 patients. Although we did not find differences in the demographics of CARDS patient compared to other study populations, an influence is conceivable. In the absence of COVID-19-specific disease severity scores, this may have resulted in the assessment of patients who were more severely affected than the general COVID-19 population.

Regarding the observed decreased need for sedation in liver or renal failure, a clear attribution to COVID-19 disease remains open. Differentiation of the extent to which the statistically significant observations are COVID-19-specific or independent of this disease is poorly clarified by our clinical study data.

Our study was able to investigate only partial aspects of the conceivable influences of restricted sedation due to the moderate sample size.

Electroencephalography (EEG) was not performed during the study to assess sedation depth.

## 6. Conclusions

Difficulties in sedation of critically ill CARDS patients have been described several times without finding optimal sedation strategies. The results of our study reveal that age and the need for vv-ECMO therapy were associated with a significant increase in sedative dosages, whereas COVID-19-associated complications such as hepatic or renal failure were associated with lower sedation doses.

## Figures and Tables

**Figure 1 jcm-11-00588-f001:**
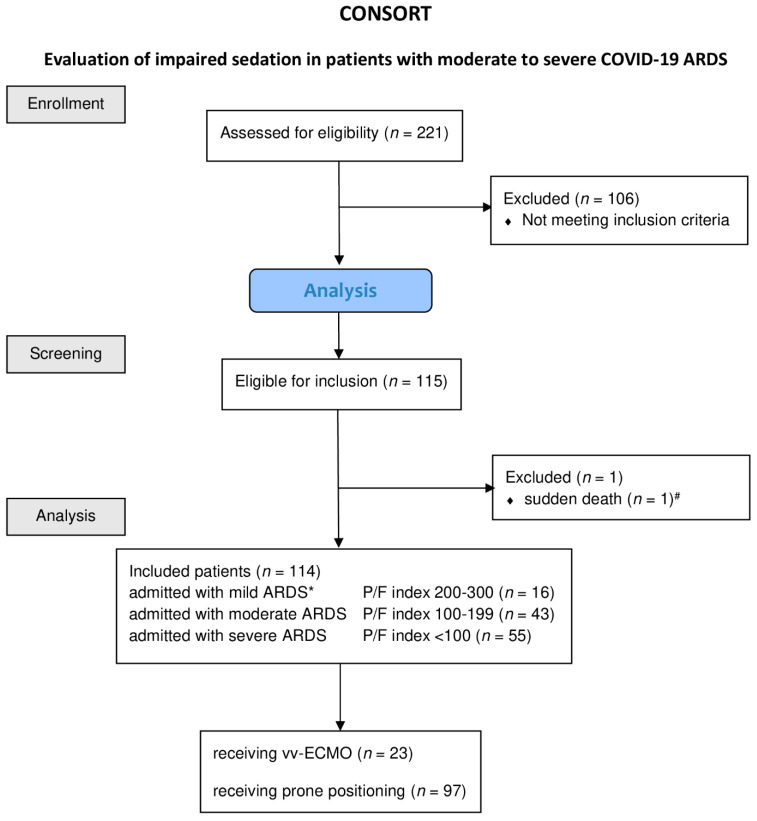
Consolidated Standards of Reporting Trials (CONSORT) diagram of patients included into the study. Diagram of the inclusion process, as well as the reasons for exclusion and treatment, patients receiving vv-ECMO or prone positioning. Abbreviations: ARDS, acute respiratory distress syndrome; paO_2_, oxygen partial pressure in arterial blood; FiO_2_, inspiratory oxygen fraction; P/F index, paO_2_·FiO_2_^−1^; vv-ECMO, veno-venous-extra corporeal membrane oxygenation. * ARDS severity classification according BERLIN definition Yes in all cases it is a minus value of the RAAS. ^#^ Patient died within less than eight hours from admission.

**Figure 2 jcm-11-00588-f002:**
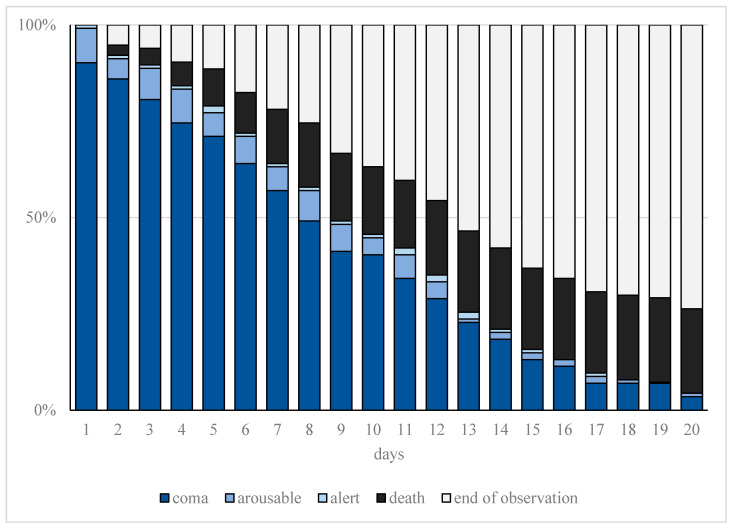
Cumulative frequency of observed sedation depth. Progression of observed sedation depth and deaths within the first 20 days of treatment in patients with moderate or severe COVID-19-related acute respiratory distress syndrome (CARDS) requiring mechanical ventilation. The representation of sedation depth (assessed on the Richmond Agitation Sedation Score (RASS)) is illustrated represented as coma (ultramarine blue, RASS ≤ −3), arousable (blue, RASS = −2), or alert (light blue, RASS ≥ −1). Furthermore, end of observation is represented for death (black) and tracheostomy or extubation (gray).

**Figure 3 jcm-11-00588-f003:**
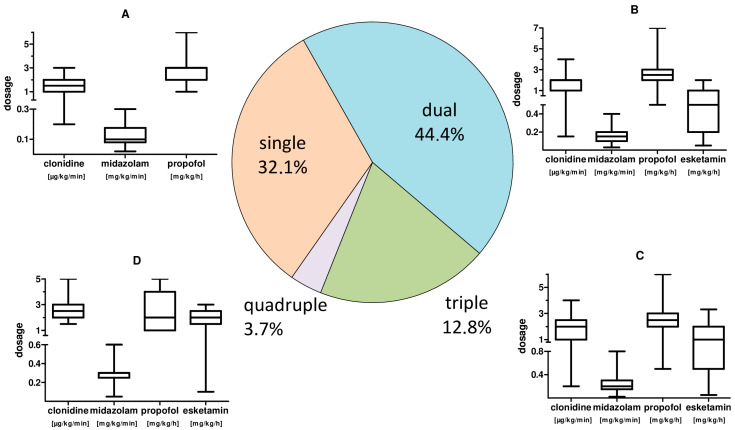
CARDS patient sedation: frequency of combined use and sedative dosages. Pie chart of the frequency of combined application of sedatives and, by analogy, the corresponding applied substance dosages. (**A**): box plot whisker plot of the sedation for single treatment. (**B**): box plot whisker plot of the individual sedation dosage for combined use of two sedatives. (**C**): box plot whisker plot of the individual sedation dosages for triple use of sedatives. (**D**): box plot whisker plot of the individual sedation dosages for quadruple use of sedatives.

**Table 1 jcm-11-00588-t001:** Clinical characteristics of CARDS patients.

Characteristics	Patients Included *n* = 114	
age, y	66	(±13.7)
sex, male	88	(77.2%)
bodyweight, kg *	93.9	(±20.7)
BMI *	30.84	(±6.7)
SAPS II *	43.88	(±10.84)
paO_2_ × FiO_2_^−1^ *	123.7	(±69.0)
median observation period, h	242.6	(±170.8)
vv-ECMO treatment	23	(20.2%)
vv-ECMO treatment time, h	277.9	(±254.2)
cRRT treatment due to AKI	43	(37.7%)
cRRT treatment time, h	189.5	(±135.2)
mortality	65	(57.0%)
coronary artery disease *	37	(32.5%)
pulmonary disease *	35	(30.7%)
diabetes *	48	(42.1%)
arterial hypertonus *	78	(68.4%)
chronic kidney disease *	24	(21.1%)
cerebrovascular events *	15	(13.2%)

Data are presented as mean (±SD) or as patient number (percentage) where applicable. Abbreviations: AKI, acute kidney injury; BMI, body mass index; cRRT, continuous renal replacement therapy; d, days; h, hours; kg, kilogram; SAPS II, Simplified Acute Physiology Score II; SD, standard deviation; vv-ECMO, veno-venous extracorporeal membrane oxygenation; y, years. * At the time of ICU admission.

**Table 2 jcm-11-00588-t002:** Linear regression for sedative dosages and CARDS associated conditions as well as preconditions.

Condition	Value	Standard Error	*p*-Value
**CARDS-associated**			
vv-ECMO therapy	0.050	0.005	<0.001
catecholamine dose	0.013	0.003	<0.001
liver failure	−0.006	0.002	0.001
renal failure	−0.008	0.003	0.001
SAPS II on admission	0.001	0.000	0.089
**Preconditions**			
age (years)	−0.001	0.000	0.003
body mass index (kg/m^2^)	0.000	0.001	0.982
coronary artery disease	−0.022	0.011	0.059
pulmonary disease	0.016	0.010	0.110
cerebrovascular disease	0.022	0.014	0.130
chronic kidney disease	−0.016	0.012	0.185
cancer	−0.025	0.023	0.275
arterial hypertonus	0.010	0.012	0.379
diabetes mellitus	−0.004	0.010	0.686
peripheral artery disease	−0.006	0.016	0.712

Results of linear regression analysis comparing sedation need in stated entity of individuals with versus without characteristic expression. Value representing the regression coefficient of the multiple regression analysis along with the estimated standard error of the stated entity on the change in sedation amount; the *p*-value indicates the statistical significance level. Abbreviations: kg/m^2^, body weight divided by height in meters in square; vv-ECMO, veno-venous extracorporeal membrane oxygenation; SAPS II, Simplified Acute Physiology Score II; CARDS, coronavirus disease 2019-induced acute respiratory distress syndrome.

## Data Availability

Data cannot be shared publicly. The datasets generated and/or analyzed during the current study are not publicly available due to national data protection laws but are available upon reasonable request from the corresponding author, or via the data protection officer of the University Hospital Frankfurt (Datenschutz@kgu.de).

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
