# Peer review of "Associated Factors of High Sedative Requirements within Patients with Moderate to Severe COVID-19 ARDS"

_jcm, 2022, doi:10.3390/jcm11030588_

Round 1

Reviewer 1 Report

In this prospective, observational, monocentric study “Associated Factors of High Sedative Requirements within Patients with Moderate to Severe COVID-19 ARDS”, Flinspach et al. evaluated the sedation requirements for COVID-19 critically ill patients, by analyzing 114 patients.

They found that COVID-19 patients require high levels of sedatives and the factor associated to influence the sedation level are age, ECMO and additional organ failure.

The paper is interesting, but the findings are not strong enough for being published in this form, especially because the low percentage of patients requiring muscle relaxant infusion needs more explanations by the authors. One of the main reasons for excessive sedation is ventilator asynchronies but the author did not provide any data about the ventilatory protocol of these patients. Only 4 /144 patients were administered with muscle relaxant. Commonly used ventilation for ARDS is controlled mechanical ventilation. if the patients were in controlled mode without any muscle relaxant, it I clear to me that the dosage of sedatives needed to suppress the respiratory effort are higher than normal. Please provide some explanation on this. In this context, the increased use sedatives could be interpreted as necessary to abolish the respiratory trigger.  Please provide comparative data between patients with and without muscle relaxant infusion (maybe adding this in the as a parameter for the linear regression). Finally, no EEG monitoring was used to assess the dept of sedation and necessity for analgesia and this has to be clearly stated in the limitations.

Specific comment:

  • line 66-68: I would move this sentence in the discussion more than in the introduction.
  • How was sedation managed in patients undergoing the infusion of muscle relaxant (RASS not reliable). Was used any kind of EEG monitoring? (eg. BIS)
  • Table1: please change Horovitz index to PaO2/FiO2 index
  • Table 1: not clear if these data are on ICU admission in the table description; please clarify.
  • Line 184: how did you define sufficient sedation?
  • Please clarify how did you assess the RASS score in prone position
  • Table 2: “civer failure”. Correct the typo.
  • Table 2: please use commas or dots for decimals.
  • One of the main reasons for sedation requirement is ventilator asynchronies but the author did not provide any data about the ventilatory protocol of these patients. Only 4 patients/144 were administered with muscle relaxant. Commonly used ventilation for ARDS is controlled mechanical ventilation. if the patients were in controlled mode without any muscle relaxant, it I clear to me that the dosage of sedatives needed to suppress the respiratory effort are higher than normal. Please provide some explanation on this.
  • Higher use of sedatives could explain why the higher incidence of PICS in these population (doi: 1007/s11136-021-02865-7) . please comment in the discussion.

Author Response

We would like to thank the reviewer for his conscientious review of our manuscript and refer to the attachment regarding point-by-point revision.

Reviewer 1:

Comments and Suggestions for Authors

In this prospective, observational, monocentric study “Associated Factors of High Sedative Requirements within Patients with Moderate to Severe COVID-19 ARDS”, Flinspach et al. evaluated the sedation requirements for COVID-19 critically ill patients, by analyzing 114 patients.

They found that COVID-19 patients require high levels of sedatives and the factor associated to influence the sedation level are age, ECMO and additional organ failure.

The paper is interesting, but the findings are not strong enough for being published in this form, especially because the low percentage of patients requiring muscle relaxant infusion needs more explanations by the authors. One of the main reasons for excessive sedation is ventilator asynchronies but the author did not provide any data about the ventilatory protocol of these patients. Only 4 /144 patients were administered with muscle relaxant. Commonly used ventilation for ARDS is controlled mechanical ventilation. if the patients were in controlled mode without any muscle relaxant, it I clear to me that the dosage of sedatives needed to suppress the respiratory effort are higher than normal. Please provide some explanation on this. In this context, the increased use sedatives could be interpreted as necessary to abolish the respiratory trigger.  Please provide comparative data between patients with and without muscle relaxant infusion (maybe adding this in the as a parameter for the linear regression).

We read the reviewer’s comments with great interest and would like to thank for these valuable comments regarding our manuscript. We acknowledge her/his concerns about increased sedation needs for achieving ventilator synchronization because of a low percentage of NMBA administration and are grateful for the opportunity to address any remaining ambiguities. For further clarification, please refer to our point by point responses below:

Finally, no EEG monitoring was used to assess the dept of sedation and necessity for analgesia and this has to be clearly stated in the limitations.

We thank the reviewer for this objection and have accordingly added this limitation to the manuscript:

"Electroencephalography (EEG) was not performed during the study to assess sedation depth."

Specific comment:

    line 66-68: I would move this sentence in the discussion more than in the introduction.

We would like to thank the reviewer and have moved the corresponding sentence in the manuscript to the discussion.

    How was sedation managed in patients undergoing the infusion of muscle relaxant (RASS not reliable). Was used any kind of EEG monitoring? (eg. BIS)

We thank you for this comment and agree that standard sedation and analgesia scores such as the Richmond Applied Agitation and Sedation Score (RASS) are not useful in the administration of muscle relaxants, nor are other scores. Therefore, it was not possible to perform a full assessment using a validated scale, but sedation and analgesia was performed based on bedside assessment and vegetative parameters when available. During the study period, EEG monitoring could not be ensured. Nevertheless, the authors are aware of the corresponding limitation and have now explicitly stated it as such due to the reviewer’s grateful hint.

    Table1: please change Horovitz index to PaO2/FiO2 index

We thank you for pointing out this ambiguity and have adapted the naming of the parameter accordingly.

    Table 1: not clear if these data are on ICU admission in the table description; please clarify.

Thank you for pointing out this ambiguity. We have adjusted our presentation correspondingly.

    Line 184: how did you define sufficient sedation?

We would like to thank the reviewer for his objection regarding the various potentially misleading terms and have standardized the reporting as requested by the reviewer.

In accordance with our standards for determining adequate sedation, a bedside examination was carried out by the attending physician in addition to an evaluation of the reliable Richmond Agitation and Sedation Scale (RASS), which assesses signs of stress and the presence of vegetative agitation (tachycardia, hypertension, sweating, tachypnea and tears in the eyes not otherwise explained).[1,2]

With regard to the target RASS, we reported different treatment regimens in the manuscript following the guidelines available to date.[3-6]

In order to make the manuscript more coherent, we adjusted the terminological differentiation to the descriptions given in lines 113-122; we have changed the term “sufficient” to “adequate”.

    Please clarify how did you assess the RASS score in prone position

We thank the reviewer for the question raised.

In order to avoid pressure ulceration, especially of the nose, the patient was placed in the prone position with the head tilted to the side. Accordingly, it was possible for the attending staff to perceive an opening of the eyes or a movement of the eyes, albeit to a limited extent.

In order to improve the comprehensibility of our manuscript, we have added a corresponding description:

"In prone position, the assessment of the RASS, in terms of eye movement and opening, was carried out by the responsible intensive care staff under limited conditions due to the head being carefully tilted to the side."

    Table 2: “civer failure”. Correct the typo.

We would like to thank the reviewer once again for the diligent review of our manuscript. We have corrected the corresponding linguistic error.

    Table 2: please use commas or dots for decimals.

We would like to thank the reviewer for his attentive observation and apologize for our mistake. We have made a correction accordingly.

    One of the main reasons for sedation requirement is ventilator asynchronies but the author did not provide any data about the ventilatory protocol of these patients. Only 4 patients/144 were administered with muscle relaxant. Commonly used ventilation for ARDS is controlled mechanical ventilation. if the patients were in controlled mode without any muscle relaxant, it I clear to me that the dosage of sedatives needed to suppress the respiratory effort are higher than normal. Please provide some explanation on this.

As the reviewer correctly states, continuous muscle relaxation was performed to a small rate in our center. However, single applications of predominantly long-acting NMBAs were also carried out in 124 cases (41 single doses of cisatracurium and 83 single doses of rocuronium). By administering these substances in combination with sedatives in the sense of a co-relaxant effect and under repeated relaxometric monitoring, relaxation was achieved in several cases in the first 48 hours of ARDS treatment. However, the use of neuromuscular blockade as a therapeutic option is still considered to have an ambiguous benefit in the context of ARDS therapy.[7,8] So far, only one application in the early phase of ARDS is recommended.[9]

Beyond that, we favor the form of spontaneous ventilation in our clinic to achieve an optimal recruitment of all lung segments. Therefore, 76.52% of the patients described in this study achieved spontaneous ventilation during the observation period. Of the remaining 23.48% of patients receiving controlled ventilation, 60.5% were on vv-ECMO therapy. The majority of these patients showed an irrelevant tidal volume (<50ml) while on ultraprotective pulmonary ventilation. Accordingly, prolonged relaxation was not necessary for ventilator synchrony.

To clarify the previous reporting (lines 312-320) we have added an expansion to the discussion:

„In our study population, due to frequent spontaneous breathing, we were mostly able to achieve adequate ventilator synchronization without NMBAs. If this could not be achieved without increasing sedation, we were able to manage this by administering single doses of NMBAs using relaxometry in the early phase of C-ARDS, with the exception of 4 cases that required continuous relaxation.”

    Higher use of sedatives could explain why the higher incidence of PICS in these population (doi: 1007/s11136-021-02865-7) . please comment in the discussion.

We thank the reviewer for this suggestion. In the context of the present work, it was not possible to report on post-intensive care syndrome. This is due to the fact that a corresponding follow-up could not be established.

However, we are pleased to meet the reviewer’s suggestion regarding the embedding of the topic of post-intensive care syndrome in relation to the increased need for sedation after C-ARDS ICU treatment in the discussion.

It has also not yet been possible to clarify the extent to which the increased sedation requirements mentioned above are related to post-intensive care syndrome (PICS). Late sequelae of severe COVID-19 infection observed in the context of PICS include not only physical but also neurological impairment and a reduced quality of life.[10,11] A link via neuroinflammation would be conceivable and should be investigated further.”

  1. Ely, E.W.; Truman, B.; Shintani, A.; Thomason, J.W.W.; Wheeler, A.P.; Gordon, S.; Francis, J.; Speroff, T.; Gautam, S.; Margolin, R.; et al. Monitoring Sedation Status Over Time in ICU PatientsReliability and Validity of the Richmond Agitation-Sedation Scale (RASS). JAMA 2003, 289, 2983-2991, doi:https://doi.org/10.1001/jama.289.22.2983.
  2. Shehabi, Y.; Bellomo, R.; Reade, M.C.; Bailey, M.; Bass, F.; Howe, B.; McArthur, C.; Murray, L.; Seppelt, I.M.; Webb, S.; et al. Early Goal-Directed Sedation Versus Standard Sedation in Mechanically Ventilated Critically Ill Patients: A Pilot Study*. Critical Care Medicine 2013, 41, 1983-1991, doi:10.1097/CCM.0b013e31828a437d.
  3. Guérin, C.; Reignier, J.; Richard, J.-C.; Beuret, P.; Gacouin, A.; Boulain, T.; Mercier, E.; Badet, M.; Mercat, A.; Baudin, O. Prone positioning in severe acute respiratory distress syndrome. New England Journal of Medicine 2013, 368, 2159-2168, doi:https://doi.org/10.1056/nejmoa1214103.
  4. Nigoghossian, C.D.; Dzierba, A.L.; Etheridge, J.; Roberts, R.; Muir, J.; Brodie, D.; Schumaker, G.; Bacchetta, M.; Ruthazer, R.; Devlin, J.W. Effect of Extracorporeal Membrane Oxygenation Use on Sedative Requirements in Patients with Severe Acute Respiratory Distress Syndrome. Pharmacotherapy 2016, 36, 607-616, doi:https://doi.org/10.1002/phar.1760.
  5. Nasraway, S.A.; Jacobi, J.; Murray, M.J.; Lumb, P.D. Sedation, analgesia, and neuromuscular blockade of the critically ill adult: revised clinical practice guidelines for 2002. Critical care medicine 2002, 30, 117-118, doi:https://doi.org/10.1097/00003246-200201000-00019.
  6. Chanques, G.; Constantin, J.-M.; Devlin, J.W.; Ely, E.W.; Fraser, G.L.; Gélinas, C.; Girard, T.D.; Guérin, C.; Jabaudon, M.; Jaber, S. Analgesia and sedation in patients with ARDS. Intensive Care Medicine 2020, 1-15, doi:https://doi.org/10.1007/s00134-020-06307-9.
  7. Gattinoni, L.; Marini, J.J. Prone positioning and neuromuscular blocking agents are part of standard care in severe ARDS patients: we are not sure. 2015, doi:https://doi.org/10.1007/s00134-015-4040-6.
  8. Ferguson, N.D.; Thompson, B.T. Prone positioning and neuromuscular blocking agents are part of standard care in severe ARDS patients: no. Intensive Care Medicine 2015, 41, 2198-2200, doi:10.1007/s00134-015-4043-3.
  9. Hraiech, S.; Forel, J.-M.; Papazian, L. The role of neuromuscular blockers in ARDS: benefits and risks. Current Opinion in Critical Care 2012, 18, 495-502, doi:https://doi.org/10.1097/mcc.0b013e328357efe1.
  10. Gamberini, L.; Mazzoli, C.A.; Sintonen, H.; Colombo, D.; Scaramuzzo, G.; Allegri, D.; Tonetti, T.; Zani, G.; Capozzi, C.; Giampalma, E.; et al. Quality of life of COVID-19 critically ill survivors after ICU discharge: 90 days follow-up. Quality of Life Research 2021, 30, 2805-2817, doi:10.1007/s11136-021-02865-7.
  11. Banno, A.; Hifumi, T.; Takahashi, Y.; Soh, M.; Sakaguchi, A.; Shimano, S.; Miyahara, Y.; Isokawa, S.; Ishii, K.; Aoki, K. One-Year Outcomes of Postintensive Care Syndrome in Critically Ill Coronavirus Disease 2019 Patients: A Single Institutional Study. Critical Care Explorations 2021, 3.

Reviewer 2 Report

I read with a great interest the study by Armin N. Flinspach and al. entiled “Associated Factors of High Sedative Requirements within Patients with Moderate to Severe COVID-19 ARDS”.

This retrospective observational study on sedation consumption in Covid-19 ARDS (CARDS), confirm a clinical observation.

Comment 1 : Drug and Alcohol abuse are a classical raison for increasing sedation, this data must be specified in table 1, despite the exclusion of this data in statistical analysis.

Comment 2 : Author report propofol as first intention drug during sedation ARDS, this is unusual practice and used in 40% of sedated patient. Midazolam or other benzodiazepine are common first intention sedative drug. When used in single therapy, adverse events of propofol were observed ?

Comment 3 : For explain the increasing of sedative drug in CARDS, we need some information on COVID-19 intensity, as percentage of lung affected, intensity of inflammatory syndrome (biological parameters as CPR or ferritin). Inflammatory syndrome could have an effect on sedation.

Comment 4 : Can authors report the incidence of neurological manifestation (anosmia, confusion, epilepsy) in COVID-19 patients before admission in ICU.

Comment 5 : In table 2 a typo : “civer failure” please correct it.

Author Response

We would like to thank the reviewer for his conscientious review of our manuscript and refer to the attachment regarding point-by-point revision.

Reviewer 2:

I read with a great interest the study by Armin N. Flinspach and al. entiled “Associated Factors of High Sedative Requirements within Patients with Moderate to Severe COVID-19 ARDS”.

This retrospective observational study on sedation consumption in Covid-19 ARDS (CARDS), confirm a clinical observation.

We read the reviewer’s comments with great interest and would like to thank for these valuable comments regarding our manuscript.

Comment 1 : Drug and Alcohol abuse are a classical raison for increasing sedation, this data must be specified in table 1, despite the exclusion of this data in statistical analysis.

According to the medical histories and laboratory tests taken on admission, no active substance abuse could be detected in the collective described by us.

There were no indications of past drug abuse in any of the patients. In two patients, there was a history of alcohol abuse that had already ceased long before their COVID-19 infection. As a result, patients with substance abuse were not included in the statistical evaluation, as the reviewer correctly indicated.

We have added a corresponding paragraph to our manuscript in order to ensure clarity in our reporting:

"In our study population, no history of drug abuse was detected. Regarding alcohol, two patients had a history of alcohol abuse. However, this was terminated prior to COVID-19 infection.”

Comment 2 : Author report propofol as first intention drug during sedation ARDS, this is unusual practice and used in 40% of sedated patient. Midazolam or other benzodiazepine are common first intention sedative drug. When used in single therapy, adverse events of propofol were observed ?

We thank the reviewer for her/his objection. However, we must clearly emphasize that the use of propofol for sedation should be considered as first-line sedation according to various recommendations for ICU sedation and C-ARDS treatment.(1-4) The frequent use of propofol instead of midazolam is due to the advantage in terms of delirium rate, which has been shown to be significantly increased by benzodiazepines.(5-7) Thus, especially at the beginning of the individual patient’s treatment and at the beginning of the pandemic, propofol was frequently used when the expected duration of ventilation was unclear.

According to the question raised regarding adverse effects, we did not find evidence of any adverse effects in the cohort observed. In particular, we did not find any laboratory evidence of Propofol Infusion Syndrome.

Comment 3 : For explain the increasing of sedative drug in CARDS, we need some information on COVID-19 intensity, as percentage of lung affected, intensity of inflammatory syndrome (biological parameters as CPR or ferritin). Inflammatory syndrome could have an effect on sedation.

We thank the reviewer for the question raised about the influence of the severity of COVID-19 infection on the need for sedation.

The severity of the infection was not determined on the basis of laboratory parameters, in particular with regard to immunomodulatory therapy, e.g. dexamethasone. At our department, interleukin-6 and procalcitonin were determined for detection of inflammation (e.g. bacterial superinfections) on a daily basis. Due to the lack of knowledge of the correlation between ferritin and severity of lung inflammation at the beginning of the pandemic, which has since been demonstrated in Lancet, among other publications, no regular examination of this parameter was carried out. The same applies to the C-reactive protein.

Regrettably, as is certainly understandable, CT imaging to quantify the lung segments still ventilated could not be carried out regularly in the most severely affected patients, for example those receiving vv-ECMO therapy. CT imaging was only carried out in exceptional cases with enormous logistical and personnel effort.

To determine the severity of C-ARDS, especially at the time of admission, but also during the course of therapy, we used the concerted BERLIN definition of ARDS as an analogy. Due to the primary lung involvement and the definition as acute respiratory distress syndrome, this seemed to be the most suitable together with the Simplified Acute Physiology Score II. In order to detect complications in the sense of extended organ involvement, we also observed liver and kidney parameters. The latter parameters were examined in the study evaluation with regard to an influence on sedation requirements.

Comment 4 : Can authors report the incidence of neurological manifestation (anosmia, confusion, epilepsy) in COVID-19 patients before admission in ICU.

We appreciate the reviewer's contribution to this interesting issue. In our patient population, we observed one patient with epilepsy prior to COVID-19 infection. This was assessed as preexisting neurological condition. The cause of other neurological symptoms (e.g. confusion, delirium) at the time of ICU-admission were difficult to differentiate. Mostly, this was due to the severity of symptoms on admission. This precluded the assessment of thorough medical history and hypoxia and infection were the most likely explanations for altered mental status. Secondly, the presence of anosmia at this critical stage of infection was either not reported or not considered relevant and therefore not regularly documented.

In order to ensure improved reporting of epilepsy, we have added a corresponding paragraph to our manuscript:

“In one patient, the medical history revealed preexisting epilepsy which had been asymptomatic for several years with therapy.“

Comment 5 : In table 2 a typo : “civer failure” please correct it.

We would like to thank the reviewer once again for the diligent review of our manuscript. We have corrected the corresponding linguistic error.

  1. Chanques G, Constantin J-M, Devlin JW, et al: Analgesia and sedation in patients with ARDS. Intensive Care Medicine 2020; 1-15
  2. Payen J-F, Chanques G, Futier E, et al: Sedation for critically ill patients with COVID-19: Which specificities? One size does not fit all. Anaesth Crit Care Pain Med 2020; S2352-5568(2320)30077-30071
  3. Hanidziar D,Bittner EA: Sedation of Mechanically Ventilated COVID-19 Patients: Challenges and Special Considerations. Anesth Analg 2020; 10.1213/ANE.0000000000004887
  4. Kapp CM, Zaeh S, Niedermeyer S, et al: The use of analgesia and sedation in mechanically ventilated patients with COVID-19 ARDS. Anesth Analg 2020;
  5. Shah FA, Girard TD,Yende S: Limiting sedation for patients with acute respiratory distress syndrome - time to wake up. Curr Opin Crit Care 2017; 23:45-51
  6. Hsieh SJ, Soto GJ, Hope AA, et al: The Association between Acute Respiratory Distress Syndrome, Delirium, and In-Hospital Mortality in Intensive Care Unit Patients. American Journal of Respiratory and Critical Care Medicine 2015; 191:71-78
  7. Zaal IJ, Devlin JW, Hazelbag M, et al: Benzodiazepine-associated delirium in critically ill adults. Intensive care medicine 2015; 41:2130-2137

Round 2

Reviewer 1 Report

I have no further comment and I would like to congratulate the authors for the effort to improve the manuscript.